# Impact of the Area of Residence of Ovarian Cancer Patients on Overall Survival

**DOI:** 10.3390/cancers14235987

**Published:** 2022-12-04

**Authors:** Floriane Jochum, Anne-Sophie Hamy, Thomas Gaillard, Lise Lecointre, Paul Gougis, Élise Dumas, Beatriz Grandal, Jean-Guillaume Feron, Enora Laas, Virginie Fourchotte, Noemie Girard, Lea Pauly, Marie Osdoit, Elodie Gauroy, Lauren Darrigues, Fabien Reyal, Cherif Akladios, Fabrice Lecuru

**Affiliations:** 1Residual Tumor & Response to Treatment Laboratory (RT2Lab), Translational Research Department, INSERM, U932 Immunity and Cancer, 75005 Paris, France; 2Department of Gynecology, Strasbourg University Hospital, 67000 Strasbourg, France; 3Department of Medical Oncology, Institut Curie, University Paris Cite, 75005 Paris, France; 4Department of Surgery, Institut Curie, University Paris Cite, 75005 Paris, France; 5ICube UMR 7357—Laboratoire des Sciences de l’Ingénieur, de l’Informatique et de l’Imagerie, Université de Strasbourg, 67000 Strasbourg, France; 6Institut Hospitalo-Universitaire (IHU), Institute for Minimally Invasive Hybrid Image-Guided Surgery, Université de Strasbourg, 67000 Strasbourg, France

**Keywords:** ovarian cancer, area of residence, hierarchical cluster algorithm, sociodemographic factor

## Abstract

**Simple Summary:**

The disparities in ovarian cancer care and outcomes have been linked to socioeconomic indicators. Our study mainly demonstrated that women living in economically and socially deprived areas had a significantly higher risk of death after adjustment for individual factors. Our results reflect the complexity of the environment in which the patients live and the impact on overall survival of the combination of negative social, economic and educational factors within that environment.

**Abstract:**

Survival disparities persist in ovarian cancer and may be linked to the environments in which patients live. The main objective of this study was to analyze the global impact of the area of residence of ovarian cancer patients on overall survival. The data were obtained from the Surveillance, Epidemiology and End Results (SEER) database. We included all the patients with epithelial ovarian cancers diagnosed between 2010 and 2016. The areas of residence were analyzed by the hierarchical clustering of the principal components to group similar counties. A multivariable Cox proportional hazards model was then fitted to evaluate the independent effect of each predictor on overall survival. We included a total of 16,806 patients. The clustering algorithm assigned the 607 counties to four clusters, with cluster 1 being the most disadvantaged and cluster 4 having the highest socioeconomic status and best access to care. The area of residence cluster remained a statistically significant independent predictor of overall survival in the multivariable analysis. The patients living in cluster 1 had a risk of death more than 25% higher than that of the patients living in cluster 4. This study highlights the importance of considering the sociodemographic factors within the patient’s area of residence when developing a care plan and follow-up.

## 1. Introduction

Ovarian cancer is the fifth leading cause of cancer-related deaths in women in the United States [1]. Despite improvements in ovarian cancer care, disparities in the quality of care and overall survival persist [2,3]. Disease onset and progression are related to social, political, ecological and historical exposures and are, therefore, closely linked to the environments in which patients live [4]. Early oncological studies exploring the effect of neighborhood socioeconomic status focused on breast cancer patients and found that residential segregation had a negative impact on all-cause mortality [5,6,7]. In economically unfavorable environments, the risk of death was more than 50% higher than that for patients from better-off areas after adjustment for age, ethnicity, sex and comorbid conditions [8]. Specifically in ovarian cancer, several studies have investigated the impact of socioeconomic status and ethnicity on ovarian cancer outcomes and treatment disparities [9,10,11]. In Bristow et al. [2], black women with ovarian cancer had a more than 30% increased risk of not receiving the recommended treatment compared to white women. Disparities in ovarian cancer care and outcomes have been linked to socioeconomic indicators, such as education level, employment status and household income, but the results remain inconsistent. In several studies, a decreased socioeconomic status has been associated with both a lower likelihood of receiving the recommended treatment and worse overall survival [2,9,10,12]. In contrast, others have failed to identify education level, income or poverty as predictive of either treatment or outcome [13]. Most of the studies performed focused on analyzing sociodemographic factors separately rather than considering the overall impact of the area in which patients live. Understanding how these sociodemographic factors interact with each other and their overall influence on ovarian cancer outcomes is critical to improving patient survival. 

The National Cancer Institute set up the Surveillance, Epidemiology and End Results (SEER) database in the United States in 1973 [14]. The SEER database is one of best-known data sources for cancer patient follow-up anywhere in the world and provides reliable data for clinical research. A key feature of this database is the inclusion of numerous sociodemographic variables for the areas of residence of the patients. The main objective of this study was to analyze the global impact of the area of residence of ovarian cancer patients on overall survival, by applying hierarchical clustering on principal components to data from the SEER database. 

## 2. Materials and Methods

### 2.1. Patient Selection

Data were obtained from the SEER database of the National Cancer Institute, the largest global open-access cancer database available, providing data from 18 population-based cancer registries covering about 28% of the total United States population [1]. The SEER database provides information about patient demographics, treatment methods, tumor characteristics, the follow-up period for survival analysis and sociodemographic data for the county of residence of the patients. We included all patients with epithelial ovarian cancer (serous, endometrioid, mucinous and clear cell) diagnosed between 1 January 2010 and 31 December 2016. Patients under the age of 18 years or diagnosed with ovarian cancer via their death certificate or at autopsy were not included. Patients with unknown American Joint Committee on Cancer (AJCC) stage or unknown tumor grade were also excluded. Finally, patients from Alaska were excluded due to the incomplete nature of sociodemographic data for this state. A flowchart of the study is provided in Figure 1.

### 2.2. Clinical and Sociodemographic Variables

The clinical features assessed in this study included age at diagnosis, ethnicity, marital status, insurance cover, tumor grade, American Joint Committee on Cancer (AJCC) stage, histological tumor type, chemotherapy, surgery, overall survival and vital status records. Age at diagnosis was analyzed both as a continuous variable and as a categorical variable with five classes (<40, 40–49, 50–59, 60–74, ≥75 years). Ethnicity was treated as a categorical variable with five classes: non-Hispanic (NH) white, NH American Indian/Alaska native, NH Asian or Pacific islander, NH black and Hispanic (all races). Marital status was classified as married, single (never married), divorced, separated, unmarried or with a domestic partner and widowed. The staging system definitions were based on the seventh edition of the AJCC staging system. For AJCC stage IV, an analysis of the different combinations of metastases was performed at diagnosis to assess their impact on overall survival and their relationship to the histological features of the tumor. In the absence of liver, lung, brain and bone metastases, patients were considered to be at stage IVA (i.e., without distant metastases). Tumor grade was classified as low (well differentiated—grade I and moderately differentiated—grade II) or high (poorly differentiated—grade III and undifferentiated—grade IV). Histological tumor types were classified according to the International Classification of Diseases for Oncology, 3rd Edition (ICD-O-3): serous (8020–8022, 8441–8442, 8460–8463, 9014), endometrioid (8380–8383, 8570), mucinous (8470–8472, 8480–8481, 9015) and clear cell (8290, 8310, 8313, 8443–8444). Finally, the 28 sociodemographic variables for each county included are listed in Appendix A. These variables relate to the following 10 fields: income/poverty, education, demographic, employment, housing, immigration, smoking, medical follow-up, rural/urban nature of the area and mobility.

### 2.3. Outcome Measurement

The primary endpoint was overall survival, defined as the time from cancer diagnosis to death from any cause or last follow-up. Data were censored for patients still alive at last follow-up. The final follow-up visit occurred on 31 December 2016.

### 2.4. Statistical Analysis

Descriptive statistics for demographic and clinical characteristics were analyzed with Chi 2 tests for categorical variables and Student’s t tests for continuous variables. To group together counties with similar characteristics, we used a clustering algorithm. Hierarchical clustering provides an excellent framework for identifying patterns and groups of similar observations in a dataset—in this case, residential areas [15]. Principal component analysis (PCA) can be applied to multidimensional datasets containing several continuous variables as a means of decreasing data dimensionality, resulting in a smaller number of variables capturing the principal information present in the initial data [16]. The combination of these two approaches, through the application of a hierarchical clustering approach to principal components, can be used to obtain a better clustering solution [17]. The first step of our model was thus to perform a PCA on the 28 county-level sociodemographic variables [17]. If the proportion of the variance explained by a dimension exceeded 15%, the dimension was retained. We then ran an agglomerative hierarchical clustering algorithm on the PCA-reduced dataset to identify areas of residence with similar sociodemographic characteristics. Ward’s method was used for cluster combination [18]. The optimal number of clusters was defined as that generating the most interpretable and best-isolated groups. All values from 2 to 6 were tested, and the results were interpreted on the basis of the sociodemographic characteristics of the groups and the differences between groups.

Survival analysis was performed by the Kaplan–Meier method, with the estimation of survival probability and log-rank tests. The univariate survival analysis was stratified for stage, with two categories: early stages (AJCC stages I and II) and advanced stages (AJCC stages III and IV). After checking the assumption of proportionality, we fitted a multivariable Cox proportional hazards model to the data to evaluate the independent effect of each predictor on survival. We searched for interactions between clusters of counties and other sociodemographic variables, such as insurance, marital status and age. The best model was selected by a stepwise top-down procedure based on the minimization of Akaike’s information criterion (AIC). Adjusted hazard ratios (HRs) and 95% confidence intervals (CIs) were generated. All statistical analyses were performed with R version 4.0.3. 

## 3. Results

### 3.1. Patient Characteristics

In total, 16,806 patients were included in this study. The characteristics of the patients are shown in Table 1. The mean age at diagnosis was 60.7 ± 13.3 years. Most patients were diagnosed with AJCC stages III–IV disease (62.6%), and serous tumors were the most prevalent histologic subtype (69.1%). The 3275 (19.5%) AJCC stage IV patients included 1968 (60.1%) patients with stage IVA tumors, 549 (16.8%) patients with liver metastases, 395 (12.1%) patients with lung metastases and 115 (3.5%) patients with combined liver and lung metastases (Figure 2A). The other 66 (2.0%) patients had bone metastases, brain metastases or rarer combinations of metastases. Metastasis data were missing for 182 (5.6%) patients. Among patients with AJCC stage IV tumors, mucinous tumors were more frequently associated with liver and lung metastases (8%) than serous (4%), endometrioid (5%) or clear-cell tumors (2%). Mucinous (13%) and clear-cell (6%) tumors were also more likely than serous (2%) and endometrioid (2%) tumors to be associated with rarer combinations of metastases (Figure 2B).

### 3.2. County Clusters

Patients from 607 different counties in four regions of the US (East, Northern Plains, Pacific Coast and Southwest) were included in this study. From the 28 county-level sociodemographic variables, we performed a PCA and selected the first two dimensions. The first and second dimensions accounted for 36% and 17% of the variance, respectively (Appendix A). The contribution of each sociodemographic variable to the dimensions is shown in Appendix A. The first dimension corresponded principally to the income and education variables. The second dimension corresponded to the immigration and demographic variables. Based on these two dimensions, the clustering algorithm assigned the 607 counties to four clusters (Figure 3). 

The characteristics of the four clusters related to education, income/poverty, employment, immigration and access to care are shown in Figure 4. The distribution of the variables related to demography, housing, smoking and mobility are presented in Appendix A. The income and education level were lowest in cluster 1 (*n* = 225 counties), the counties of which had a low proportion of immigrants, were rather rural in nature and had low levels of access to healthcare. Cluster 2 (*n* = 67 counties) also corresponded to socially deprived areas, with low income and education levels, a high proportion of foreigners or immigrants and low levels of access to healthcare, but this cluster was more urban than cluster 1. Cluster 3 (*n* = 228 counties) corresponded to well-off areas with a high standard of living and high education levels, often rural, with a low proportion of immigrants and good access to healthcare. Finally, cluster 4 (*n* = 87 counties), like cluster 3, corresponded to well-off areas but mostly in urban settings and with a more diverse population. Access to healthcare was the greatest in cluster 4, in which 79.4% of the population had undergone a Pap smear test within the last three years, versus 72.4% in cluster 1 (*p* < 0.001). A map representing the distribution of the different clusters in the United States is shown in Figure 5. 

The clinical and pathological characteristics of the patients are shown by county cluster in Table 1. The patients from cluster 1 and 2 were diagnosed in advanced stages in 61.6% of the cases versus 62.5% in clusters 3 and 4. Among the patients from cluster 1 with AJCC stage IV tumors, 96 (59.6%) were diagnosed with stage IVA and 65 (40.4%) with stage IVB, including 29 (18.0%) patients with isolated lung metastases. In cluster 4, 948 (63.6%) patients were diagnosed with stage IVA and 542 (36.4%) with stage IVB, of which 192 (12.9%) had isolated lung metastases (Appendix A, *p* < 0.001). 

### 3.3. Survival Analysis

A univariable survival analysis revealed a significant difference in the overall survival between the patients from different county clusters for AJCC stages III–IV but not for AJCC stages I–II (Figure 6a,b). 

The five-year overall survival for the patients with early cancer was 78.0% for cluster 1 and 83.0% for cluster 4. The five-year overall survival rates for the patients with advanced-stage cancers were 29.5% for cluster 1 and 39.6% for cluster 4. The median overall survival for patients with AJCC stage III–IV tumors was 37 months for cluster 1 and 45 months for cluster 4. The median follow-up was 30.0 (range: 16.0–50.0) months.

A multivariable survival analysis confirmed the known negative effects on the prognosis of older age, higher AJCC stage, particularly in cases of associated metastases, being black, having a mucinous or clear-cell tumor, high-grade tumors and, finally, a lack of surgery and/or chemotherapy (Figure 7). No interaction between the clusters and the other variables was found. After adjustment for other factors, the county cluster remained a statistically significant and independent predictor of overall survival. The patients from cluster 1 had a risk of death more than 25% higher than that for the patients from cluster 4 (HR 1.3, 95% CI 1.1 to 1.4, *p* < 0.001).

## 4. Discussion

In this study, we analyzed the combined effect of the sociodemographic characteristics of the patient’s area of residence on the overall survival in patients with epithelial ovarian cancer. We found significant disparities in survival between counties defined as disadvantaged or well-off in terms of social and economic factors, education and access to care. In particular, women living in economically and socially deprived areas were found to have a significantly higher risk of death after adjustment for individual factors. 

The major impact of socioeconomic inequalities on the survival of cancer patients has already been documented elsewhere [4]. Survival is consistently poorer in cancer patients of low socioeconomic status than in those of higher socioeconomic status, whether individual-level or geographic measures are used [4,19,20]. Studies on ovarian cancer have shown mortality to be higher, after adjustment for several individual factors, in unmarried women [21] and in women without health insurance [22]. Our study generated similar results. Ethnic differences were also confirmed in our study, in which overall survival was worst for non-Hispanic black women [2]. Published findings have also reported a negative impact of a poor neighborhood [9] or economic and racial residential segregation [12] on overall survival in patients with ovarian cancer. Our results, capturing the complexity of the environment in which the patients live and the impact on overall survival of the combination of negative social, economic and educational factors within that environment, confirm these previous findings.

Little is known about the mechanisms underlying the observed relationship between the area of residence and health, but several hypotheses can be put forward. One of the hypotheses frequently proposed in previous studies is that women with a lower socioeconomic status seek treatment late or are not directly referred to an expert center [23,24]. In women living in areas of low socioeconomic standing, a cancer diagnosis is often delayed, resulting in a decrease in survival. However, the symptoms of ovarian cancer are often mild or non-specific, and about 70% of cases are diagnosed at an advanced stage, even in well-off areas. Organized screening for ovarian cancer has not proved effective, and no screening test has yet been approved for the early diagnosis of ovarian cancer. In our study, patients were diagnosed at an advanced stage in 61.6% of cases in clusters 1 and 2 versus 62.5% in clusters 3 and 4. Comorbidity levels were also higher in cancer patients with a low socioeconomic status [25], increasing all-cause mortality [26,27]. The presence of multiple comorbid conditions may influence the time to diagnosis by leading to a delay in seeking care or the attribution of the symptoms to an existing secondary disease. Multiple comorbid conditions may also influence the choice or aggressiveness of cancer treatment in the face of a low performance status at diagnosis or contraindications for treatment in the cases of comedication [28]. Other studies have suggested that geographic variations in ovarian cancer-specific survival may be due to important predictors, such as receiving care in accordance with guidelines. By contrast to the situation for women with ovarian cancer in the general population, a British study of 1,406 ovarian cancer patients enrolled in two randomized controlled trials found no socioeconomic inequalities in ovarian cancer survival [29]. The authors concluded that the persistent socioeconomic gap in survival in the general population may be due to differential access to treatment and standards of care. Patients in areas of lower socioeconomic status tend to receive suboptimal care [2,23]. In our study, 31 (3.8%) patients in cluster 1 did not undergo surgery versus 243 (2.9%) in cluster 4 (*p* = 0.02). A similar disparity was observed for chemotherapy, with 205 (25.3%) patients in cluster 1 not receiving chemotherapy versus 1,699 (20.4%) in cluster 4. Overall, improving access to expert care and ensuring that guideline-adherent treatment is administered should be priorities in the optimization of survival in patients with ovarian cancer [30]. Most underprivileged areas are rural, and the distance to a hospital or an expert center may itself constitute an obstacle to seeking treatment or to receiving guideline-adherent care [31].

## 5. Conclusions

This study highlights the importance of considering the sociodemographic factors within the patient’s area of residence when developing a care plan and follow-up. Improvements in the management of comorbid conditions, the provision of better services for patients of low socioeconomic status and the centralization of care are immediate actions that could be taken to ensure optimal care for all. Future studies of ovarian cancer should consider the area of residence of the patients and should develop analytical and statistical approaches appropriate for the geospatial and multilevel nature of the data. Incorporating social and environmental factors into ovarian cancer etiology and outcome research should help to improve our understanding of disease processes and the identification of vulnerable populations and should make it possible to generate outcomes relevant to the entire affected population.

## Figures and Tables

**Figure 1 cancers-14-05987-f001:**
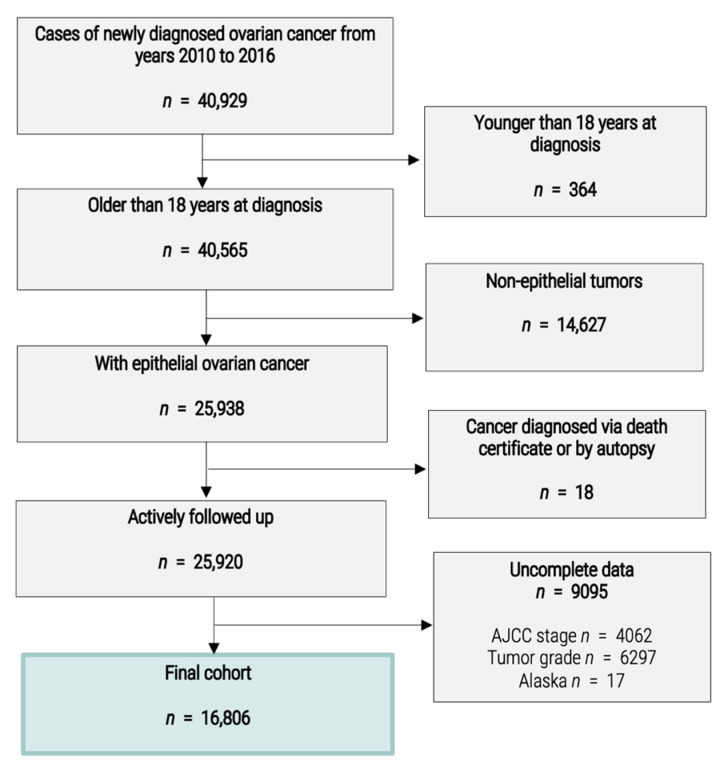
Flowchart.

**Figure 2 cancers-14-05987-f002:**
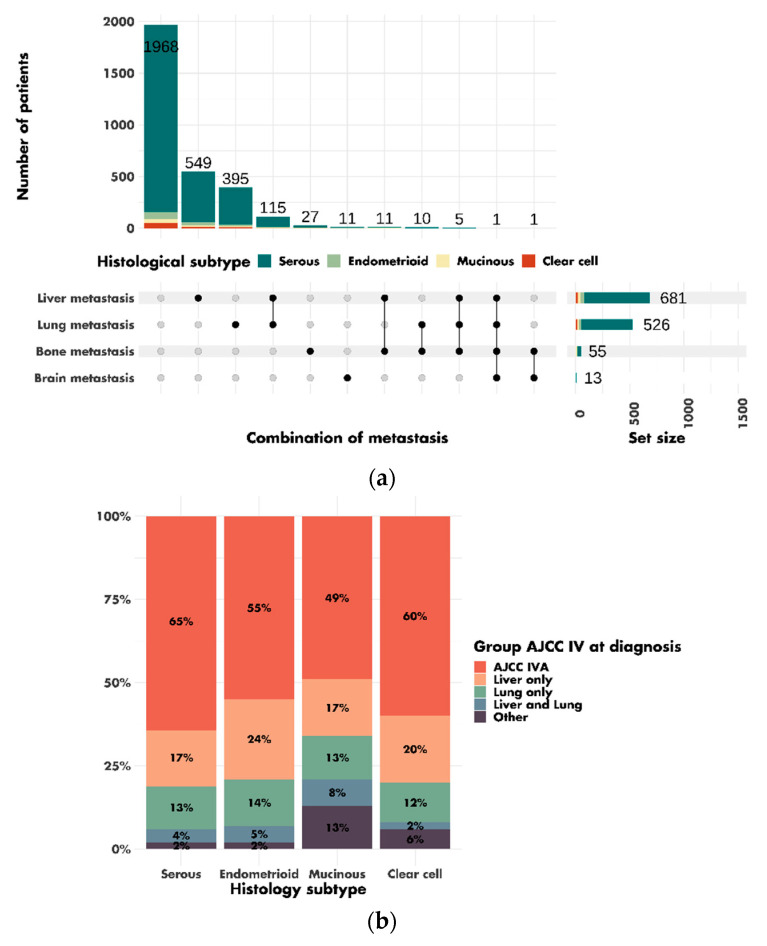
Combinations of metastases in patients with AJCC stage IV tumors at diagnosis: (**a**) description of the different combinations of metastases; (**b**) distribution of associations of metastases by histological subtype (*p* < 0.001).

**Figure 3 cancers-14-05987-f003:**
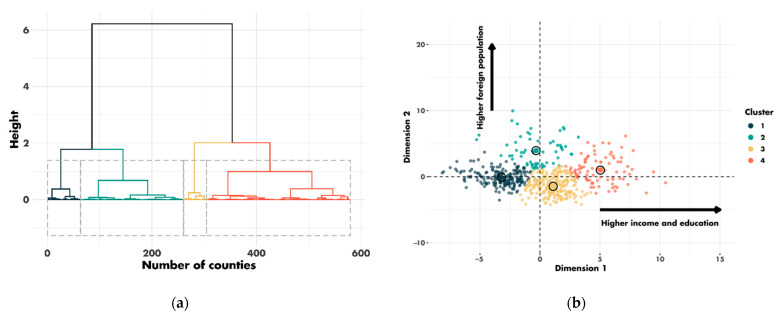
Clustering results: (**a**) cluster dendrogram; (**b**) distribution of county clusters according to the first two dimensions.

**Figure 4 cancers-14-05987-f004:**
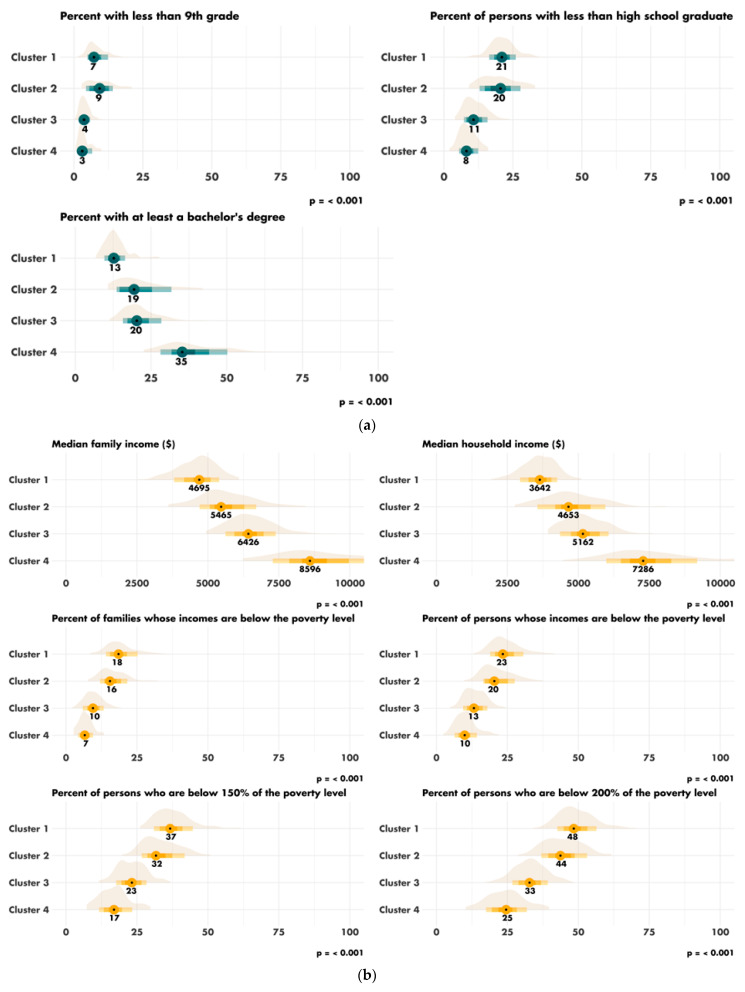
Characteristics of the clusters: (**a**) distribution of the variables related to education; (**b**) distribution of the variables related to income and poverty; (**c**) distribution of the variable related to employment; (**d**) distribution of the variables related to immigration; (**e**) disparity in access to care between the four clusters. Access to care is most difficult in cluster 1 and easiest in cluster 4. FOBT = Fecal Occult Blood Test. CRC = Colorectal Cancer Screening Blood Test. (**f**) Rural/urban distribution of the population in each cluster.

**Figure 5 cancers-14-05987-f005:**
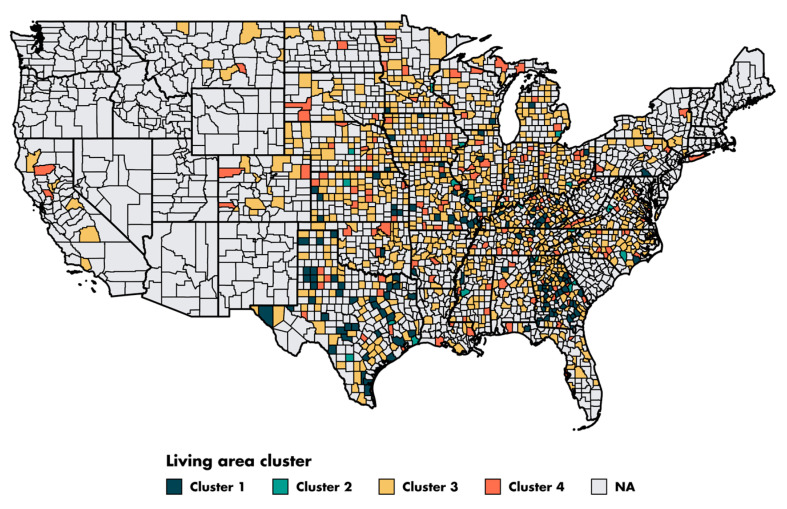
Map of the county clusters in the United States. Of the 3143 counties in the United States, 609 are represented in the ovarian cancer SEER. We excluded two counties located in Alaska due to a lack of data.

**Figure 6 cancers-14-05987-f006:**
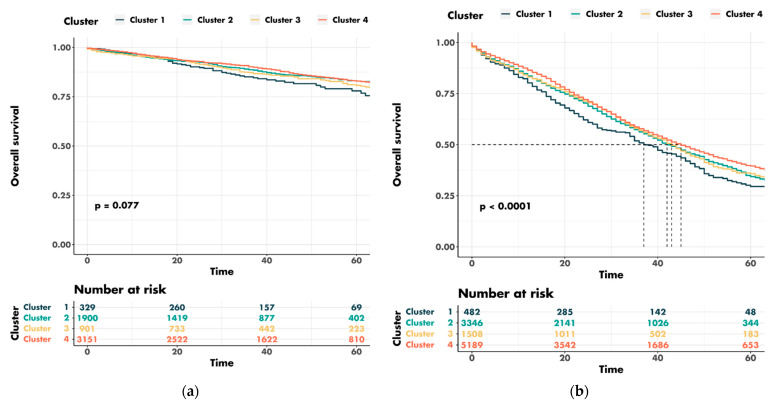
Univariable survival analysis for county clusters stratified for cancer stage: (**a**) early stages; (**b**) advanced stages.

**Figure 7 cancers-14-05987-f007:**
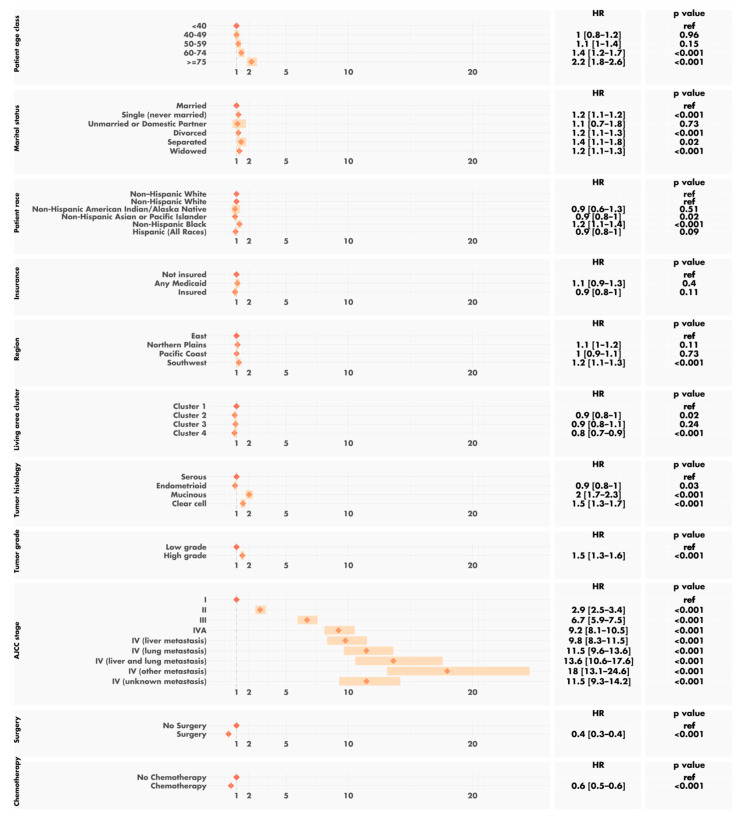
Multivariable survival analysis.

**Table 1 cancers-14-05987-t001:** Patient characteristics.

Variable		Overall	Cluster 1	Cluster 2	Cluster 3	Cluster 4	*p*
	*n*	16,806	811 (4.8)	5246 (31.2)	2409 (14.3)	8340 (49.6)	
Patient age (y)		60.7 (13.0)	61.5 (12.9)	60.0 (13.1)	61.9 (12.8)	60.8 (12.9)	<0.001
Patient age class (y)	<40	853 (5.1)	46 (5.7)	305 (5.8)	104 (4.3)	398 (4.8)	<0.001
	40–49	2312 (13.8)	90 (11.1)	771 (14.7)	286 (11.9)	1165 (14.0)	
	50–59	4564 (27.2)	203 (25.0)	1453 (27.7)	604 (25.1)	2304 (27.6)	
	60–74	6558 (39.0)	349 (43.0)	1975 (37.6)	1020 (42.3)	3214 (38.5)	
	≥75	2519 (15.0)	123 (15.2)	742 (14.1)	395 (16.4)	1259 (15.1)	
Ethnicity	NH white	11,866 (70.6)	665 (82.0)	2916 (55.6)	2118 (87.9)	6167 (73.9)	<0.001
	NH American Indian/Alaska native	96 (0.6)	5 (0.6)	48 (0.9)	13 (0.5)	30 (0.4)	
	NH Asian or Pacific islander	1564 (9.3)	7 (0.9)	510 (9.7)	45 (1.9)	1002 (12.0)	
	NH black	1105 (6.6)	98 (12.1)	481 (9.2)	138 (5.7)	388 (4.7)	
	Hispanic (All races)	2136 (12.7)	35 (4.3)	1278 (24.4)	94 (3.9)	729 (8.7)	
Marital status	Married	8950 (53.3)	428 (52.8)	2594 (49.4)	1354 (56.2)	4574 (54.8)	<0.001
	Single (never married)	3230 (19.2)	125 (15.4)	1226 (23.4)	353 (14.7)	1526 (18.3)	
	Unmarried or living with a partner	73 (0.4)	2 (0.2)	25 (0.5)	4 (0.2)	42 (0.5)	
	Divorced	1731 (10.3)	76 (9.4)	523 (10.0)	239 (9.9)	893 (10.7)	
	Separated	165 (1.0)	7 (0.9)	76 (1.4)	18 (0.7)	64 (0.8)	
	Widowed	1960 (11.7)	137 (16.9)	583 (11.1)	329 (13.7)	911 (10.9)	
	Missing	697 (4.1)	36 (4.4)	219 (4.2)	112 (4.6)	330 (4.0)	
Health insurance	Uninsured	566 (3.4)	53 (6.5)	203 (3.9)	107 (4.4)	203 (2.4)	<0.001
	Any Medicaid	1922 (11.4)	142 (17.5)	863 (16.5)	211 (8.8)	706 (8.5)	
	Insured	14,120 (84.0)	607 (74.8)	4124 (78.6)	2056 (85.3)	7333 (87.9)	
	Missing	198 (1.2)	9 (1.1)	56 (1.1)	35 (1.5)	98 (1.2)	
Region	East	5768 (34.3)	707 (87.2)	810 (15.4)	1355 (56.2)	2896 (34.7)	<0.001
	Northern Plains	1470 (8.7)	0 (0.0)	362 (6.9)	579 (24.0)	529 (6.3)	
	Pacific Coast	8650 (51.5)	40 (4.9)	3787 (72.2)	360 (14.9)	4463 (53.5)	
	Southwest	918 (5.5)	64 (7.9)	287 (5.5)	115 (4.8)	452 (5.4)	
Tumor histology	Serous	11,613 (69.1)	569 (70.2)	3575 (68.1)	1696 (70.4)	5773 (69.2)	<0.001
	Endometrioid	2699 (16.1)	134 (16.5)	905 (17.3)	355 (14.7)	1305 (15.6)	
	Mucinous	1236 (7.4)	76 (9.4)	385 (7.3)	195 (8.1)	580 (7.0)	
	Clear cell	1258 (7.5)	32 (3.9)	381 (7.3)	163 (6.8)	682 (8.2)	
AJCC stage	I	4537 (27.0)	225 (27.7)	1388 (26.5)	636 (26.4)	2288 (27.4)	<0.001
	II	1744 (10.4)	104 (12.8)	512 (9.8)	265 (11.0)	863 (10.3)	
	III	7250 (43.1)	328 (40.4)	2193 (41.8)	1108 (46.0)	3621 (43.4)	
	IVA	1968 (11.7)	88 (10.9)	676 (12.9)	254 (10.5)	950 (11.4)	
	IV (liver metastasis)	549 (3.3)	27 (3.3)	202 (3.9)	57 (2.4)	263 (3.2)	
	IV (lung metastasis)	395 (2.4)	27 (3.3)	128 (2.4)	52 (2.2)	188 (2.3)	
	IV (liver and lung metastases)	115 (0.7)	4 (0.5)	40 (0.8)	12 (0.5)	59 (0.7)	
	IV (other metastases)	66 (0.4)	3 (0.4)	23 (0.4)	5 (0.2)	35 (0.4)	
	IV (unknown metastases)	182 (1.1)	5 (0.6)	84 (1.6)	20 (0.8)	73 (0.9)	
Tumor grade	Low grade	4697 (27.9)	249 (30.7)	1519 (29.0)	704 (29.2)	2225 (26.7)	0.003
	High grade	12,109 (72.1)	562 (69.3)	3727 (71.0)	1705 (70.8)	6115 (73.3)	
Surgery	No surgery	538 (3.2)	31 (3.8)	188 (3.6)	76 (3.2)	243 (2.9)	0.020
	Surgery	16,245 (96.7)	777 (95.8)	5052 (96.3)	2326 (96.6)	8090 (97.0)	
	Missing	23 (0.1)	3 (0.4)	6 (0.1)	7 (0.3)	7 (0.1)	
Chemotherapy	No chemotherapy/Unknown	3849 (22.9)	205 (25.3)	1422 (27.1)	523 (21.7)	1699 (20.4)	<0.001
	Chemotherapy	12,957 (77.1)	606 (74.7)	3824 (72.9)	1886 (78.3)	6641 (79.6)	

NH: non-Hispanic.

## Data Availability

The data on which this article is based are available from the National Cancer Institute’s Surveillance, Epidemiology, and End Results database at: https://seer.cancer.gov/, accessed on 21 February 2022.

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
