# Peer review of "Impact of the Area of Residence of Ovarian Cancer Patients on Overall Survival"

_cancers, 2022, doi:10.3390/cancers14235987_

Round 1

Reviewer 1 Report

The authors give a very important and unique perspective on the effect of location/environment on ovarian cancer survival. I admire their work and ingenuity. I have the following comments:

1)for all figures, please increase the font size (including numbers and the texts on the axis (too hard to see and very small)

2) For figure 4, the figures seem randomly organized. Why is smoking lumped with having a mammogram? If you are going to use this data, please organize it according to social status, economic status, health etc and group them into these categories accordingly. Your data is precious but lumping them randomly together like this dirties the entire paper. When organized well, it shines, so please reconsider the organization 

3) Figure 6b needs to have dotted lines in the figure just as was done for 6a

4) again as in comment 2) please reorganize accordingly. category "insurance" should not be lumped with other aspect that are completely unrelated. I would suggest color coding these so that for example "economic" is green, "social" is red and "health" is blue etc...

No further comments. 

Author Response

Point 1: The authors give a very important and unique perspective on the effect of location/environment on ovarian cancer survival. I admire their work and ingenuity. I have the following comments:

For all figures, please increase the font size (including numbers and the texts on the axis (too hard to see and very small).

Response 1: Thank you very much for your review and your comment. The font size have been increased for all figures. 

Point 2: For figure 4, the figures seem randomly organized. Why is smoking lumped with having a mammogram? If you are going to use this data, please organize it according to social status, economic status, health etc and group them into these categories accordingly. Your data is precious but lumping them randomly together like this dirties the entire paper. When organized well, it shines, so please reconsider the organization.

Response 2: Thank you very much for your comment. The figures have been modified and separated by category. To improve the clarity of the results, a color has been defined for each category. Figures representing the distribution of variables related to education, income/poverty, employment, immigration, and access to care have been included in the main manuscript (Figure 4). The distribution of variables related to demography, housing, smoking, and mobility are shown in the supplementary material (Figures S3 to S6).

Point 3: Figure 6b needs to have dotted lines in the figure just as was done for 6a.

Response 3: Median survival was not achieved for early stages (6a). The addition of the dotted lines is therefore unfortunately not possible.

Point 4: Again as in comment 2) please reorganize accordingly. category "insurance" should not be lumped with other aspect that are completely unrelated. I would suggest color coding these so that for example "economic" is green, "social" is red and "health" is blue etc...

Response 4: Thank you again for your comment. As discussed in Response 2, the figures representing the sociodemographic variables were reorganized and separated by category. A color was chosen for each category to improve the clarity.

Reviewer 2 Report

Excellent paper. One suggestion for clarification-in looking at the map of the clusters, there are large areas of the Western US that have no clusters. Can the reason for that be clarified in the paper? Thank you.

Author Response

Point 1: Excellent paper. One suggestion for clarification-in looking at the map of the clusters, there are large areas of the Western US that have no clusters. Can the reason for that be clarified in the paper? Thank you.

Response 1: Thank you very much for your review and your comment. The Surveillance, Epidemiology, and End Results (SEER) program, originally funded by the National Cancer Institute (NCI) in 1973, collects information of cancer incidence and survival from strategically selected cancer registries of the USA. The SEER program began by including nine registries and has expanded over time to include 18 registries. A sentence about this is present line 77: “Data were obtained from the SEER database of the National Cancer Institute, the largest global open-access cancer database available, providing data from 18 population-based cancer registries covering about 28% of the total United States population”. Currently data from 609 of 3,143 counties are available in the ovarian cancer SEER. We excluded two counties located in Alaska due to lack of data. In total, 607 counties were included in our study. An annotation to Figure 5 has been added to clarify missing clusters: "Of the 3,143 counties in the United States, 609 are represented in the ovarian cancer SEER. We excluded two counties located in Alaska due to lack of data.”

Reviewer 3 Report

Jochum et al. present an interesting manuscript that examines the association between disparities in ovarian cancer care (as measured by area of residence) and overall survival. The study examined overall survival in 16,806 ovarian cancer patients from the SEER database including patients from 607 counties in four regions of the US.  They demonstrated that patients living in counties having the highest socioeconomic status and access to care, had significantly improved overall survival compared to patients living in disadvantaged counties, even after adjustment for individual factors.  These findings highlight the importance for considering the sociodemographics of the area in which a patient lives when developing a care plan for them.

This was a very interesting manuscript that was well written. In their examination of the relationship between residential socioeconomic factors and overall survival of ovarian cancer patients, the authors have identified an important barrier to equitable healthcare.  The cohort examined was well characterized including clear, reasonable inclusion and exclusion criteria.  The discussion was clear and provided interesting, evidence-based points examining possible explanations for the observed relationship between survival and area of residence.

In reading the manuscript, there were few critiques or concerns noted.  However, I did have a question regarding the prevalence of ovarian cancer in patients residing in counties with sociodemographic variables that align with cluster 1.  Cluster 1 appears to represent one of the largest groups with respect to the number of counties (225/607 counties).  However, despite this there are only 811/16806 (4.8%) cohort patients that belong to this cluster.  This is a considerable smaller number in this cluster compared to the other clusters.  Is the prevalence of ovarian cancer predicted to be similar across all four clusters?  If the prevalence of ovarian cancer is different in cluster 1 compared to the other clusters, is there information available to explain these differences?

The only other minor comment that I had was with respect to the size of font in the figures provided.  In several figures the font was very small making it difficult to read.

In general, I thought that this manuscript was interesting to read and added important information to the literature with regards to factors influencing overall survival in ovarian cancer patients.

Thank you for the opportunity to review this interesting manuscript.

Author Response

Point 1: In reading the manuscript, there were few critiques or concerns noted.  However, I did have a question regarding the prevalence of ovarian cancer in patients residing in counties with sociodemographic variables that align with cluster 1.  Cluster 1 appears to represent one of the largest groups with respect to the number of counties (225/607 counties).  However, despite this, there are only 811/16806 (4.8%) cohort patients that belong to this cluster.  This is a considerable smaller number in this cluster compared to the other clusters.  Is the prevalence of ovarian cancer predicted to be similar across all four clusters?  If the prevalence of ovarian cancer is different in cluster 1 compared to the other clusters, is there information available to explain these differences?

Response 1: Thank you very much for your review and your comment. Cluster 1 represents 225 counties out of the 607, but only 4.8% of the included patients are from this cluster. This is mainly related to the fact that cluster 1 is composed of mostly rural counties with a very low population density (Figure 6f). Cluster 3, also predominantly rural, represents 228 counties out of 607. Only 13% of the patients included in our study belong to this cluster. The prevalence is assumed to be the same between the four clusters, but unfortunately, this cannot be confirmed in our study.

Point 2: The only other minor comment that I had was with respect to the size of font in the figures provided.  In several figures the font was very small making it difficult to read.

Response 2: Thank you very much for your comment. The font size have been increased for all figures.